# KMD clustering: robust general-purpose clustering of biological data

Aviv Zelig[1,2,3], Hagai Kariti [2,3] & Noam Kaplan [2✉]

The noisy and high-dimensional nature of biological data has spawned advanced clustering algorithms that are tailored for specific biological datatypes. However, the performance of such methods varies greatly between datasets and they require post hoc tuning of cryptic hyperparameters. We present k minimal distance (KMD) clustering, a general-purpose method based on a generalization of single and average linkage hierarchical clustering. We introduce a generalized silhouette-like function to eliminate the cryptic hyperparameter k, and use sampling to enable application to million-object datasets. Rigorous comparisons to general and specialized clustering methods on simulated, mass cytometry and scRNA-seq datasets show consistent high performance of KMD clustering across all datasets.

[1] Data Science & Engineering Program, Faculty of Industrial Engineering & Management, Technion - Israel Institute of Technology, Haifa, Israel. [2] Department of Physiology, Biophysics & Systems Biology, Rappaport Faculty of Medicine, Technion – Israel Institute of Technology, Haifa, Israel. [3] These authors contributed equally: Aviv Zelig, Hagai Kariti. ✉email: noam.kaplan@technion.ac.il

Clustering is a ubiquitous set of machine learning techniques that are widely used in data analysis to computationally group sets of objects based on some measure of pairwise distance or similarity. The application of clustering to complex biological datasets is widespread[1–6], and some notable recent examples include the clustering of single cell RNA sequencing[7] (scRNA-seq) and mass cytometry data[8] with the aim of detecting cell subpopulations. In complex biological applications, under-performance of standard general-purpose clustering algorithms on noisy high-dimensional data has led to the development of clustering algorithms that specialize in specific subtypes of biological data. While these algorithms outperform general-purpose clustering algorithms on these datasets, they are often not transferrable to other types of data and their performance can vary significantly even on datasets of the same type[9,10]. Several examples of such specialized clustering methods exist, based on a variety of approaches. In scRNA-seq clustering of cells, SCCAF[11] uses a self-projection machine learning approach on a pre-clustered dataset to simultaneously identify distinct cell groups and a weighted list of feature genes for each group. In mass cytometry, FlowSOM[12] clusters the nodes of a constructed self organizing map (SOM) connected by a minimal spanning tree using consensus hierarchical clustering. One exception to these specialized methods is PhenoGraph[13] (other implementations include Scanpy[14] Louvain and Seurat[15] Louvain and Bluster[16] Louvain), which uses a shared neighbor graph to perform Louvain community detection and can be used on both types of biological datasets.

An important issue with many modern clustering methods, including the aforementioned specialized clustering algorithms, is the requirement of user-specified numerical hyperparameters. Although the values of such hyperparameters can change the clustering results dramatically[1], they are often cryptic, in the sense that they do not have a sufficiently clear interpretation in the context of the biological problem such that a user would be able to set them a priori based on biological knowledge. Ultimately this leaves users to either use inadequate default hyperparameter settings or to adjust the hyperparameter values until they are happy with the outcome, which may lead to undetectably biased results and overfitting. For example, SCCAF

is dependent on the self-projection machine learning parameters, clustering parameters and minimum self-projection accuracy parameter[11]; FlowSOM has several parameters regarding starting population, clustering channels and SOM settings[12]; PhenoGraph parameters include the number of nearest neighbors when constructing the cell contact graph[13]. Ideally, one would like a clustering algorithm to either not have such hyperparameters or have hyperparameters that are easy to set (e.g. interpretable and/or do not significantly affect the results).

Here we present a new general-purpose clustering method which we call k-minimal distances (KMD) clustering. Our method is based on a natural generalization of average and single linkage, within the framework of hierarchical clustering. Building on this generalized linkage function, we demonstrate its efficient computation and combine it with a simple outlier-aware partitioning scheme in order to improve performance in noisy scenarios. Next, we propose a generalized silhouette-like function which matches our generalized linkage function, and show that it is predictive of clustering performance across different values of the hyperparameter $k$. Using this, we are able to eliminate the need to choose a value for the cryptic hyperparameter $k$. We then apply our method to a range of simulated and experimental biological datasets in which ground truth is known and find that our method compares favorably to both standard general-purpose algorithms and domain-specific state-of-the-art algorithms. Finally, we show how a sampling-based extension of our approach can be used to apply our method to very large datasets on the scale of millions.

## Results

**KMD linkage**. We sought to generalize the notion of single and average linkage within the framework of agglomerative hierarchical clustering, in order to combine the most useful properties of both linkage types (Fig. 1a). Agglomerative hierarchical clustering starts with $n$ clusters of size one and then iteratively merges the two nearest (most similar) clusters, based on a $n$x$n$ distance matrix specifying the distance between each pair of objects. Since only distance between objects is given, a key step is to define how to calculate the distances between any two clusters, and this

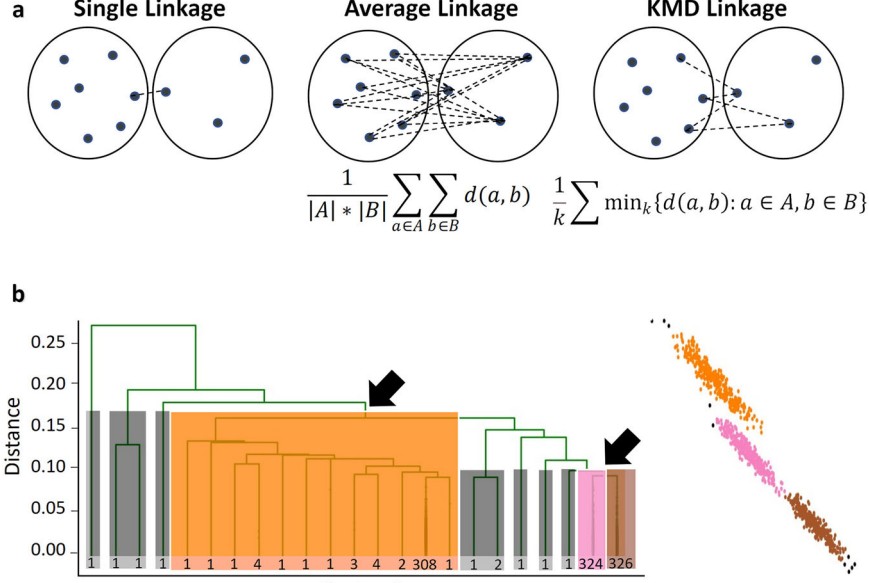

**Fig. 1 KMD clustering. a** Illustration of single, average and KMD linkage methods. **b** Outlier-aware partitioning schematic example (Left: dendrogram; Right: data points). Orange, pink and brown indicate core clusters; grey rectangles/points represent outliers; black arrows indicate merges between core clusters.

function is known as *linkage*. In *single linkage*, the distance between two clusters is defined as the minimal pairwise inter-cluster distance. Single linkage can detect both globular and complex non-globular cluster shapes, but is highly prone to noise due to its reliance on a single pairwise distance[17]. In *average linkage*, the distance between two clusters is defined as the average of all pairwise inter-cluster distances[18,19]. Average linkage clustering is more robust to noise than single linkage[17], but is biased towards detecting globular clusters[19]. We propose a generalized form of linkage which we refer to as *k minimal distances linkage* (*KMD linkage*), in which we define the distance between two clusters as the average of the *k* minimal pairwise inter-cluster distances, where *k* is an integer (Fig. 1a). Note that $k = 1$ gives single linkage and $k \gg n$ gives average linkage. Thus, we hypothesized that an intermediate value of *k* might potentially capture the best features of both worlds: be more robust to noise than single linkage, yet less biased towards globular clusters than average linkage. As we show later, prior knowledge of the value of *k* is not needed since *k* can be estimated computationally. While the implementation of a hierarchical clustering algorithm with an arbitrary linkage function can be computationally impractical, efficient algorithms for single and average linkage clustering are known. Similarly, for KMD linkage we show a method for efficient linkage update in linear time, as well as a quadratic memory implementation which is importantly independent of *k*, making computations with large *k* feasible (see Methods).

**Outlier-aware partitioning**. In the formulation of clustering that we address, the number of clusters *c* is prespecified by the user (see Discussion). In order to partition the data into *c* clusters in standard single and average linkage clustering, the standard approach is to stop the clustering (merging) process when *c* clusters are left[20]. However, in noisy datasets where an unknown number of outlier objects may exist in addition to the *c* clusters, outliers will often be merged towards the end of the clustering process (because they are far from other objects), leading to incorrect merging of large clusters under the standard partitioning approach.

To address this challenge, we implemented a simple size-based outlier-aware partitioning scheme (Fig. 1b), akin to the approach proposed in HDBSCAN[21]. We assumed that tiny clusters (smaller than a prespecified size threshold) which are merged in the final stages of the partitioning are likely outliers. Thus, our scheme ignores such merges in the final clustering steps and maintains only the last *c*-1 merges between clusters larger than the size threshold. Clusters that are smaller than the size threshold are treated separately as outliers, and the remaining *c* clusters are considered as *core clusters*. Although the outlier cluster size threshold is set by the user, it is interpretable and should be set to a value larger than a cluster size that the user considers to be too small to be meaningful and below the minimal expected cluster size. Additionally, we observe that clustering performance is typically robust across a large range of values for this threshold (Fig. S1, see "Evaluation on simulated datasets" for details on datasets), so prior knowledge of its exact value may not be required.

As it still may be useful to associate the outliers to one of the core clusters after the initial clustering process, especially when comparing predictive performance of different methods, it is straightforward to assign outliers to core clusters by simply calculating the KMD linkage of an outlier to each of the core clusters and assigning it to the core cluster with the minimal distance. In the interest of fairness, all comparisons of clustering performance in this paper were done using these outlier assignments, such that no objects were left out unless explicitly stated otherwise. Finally, we assign each outlier classification a

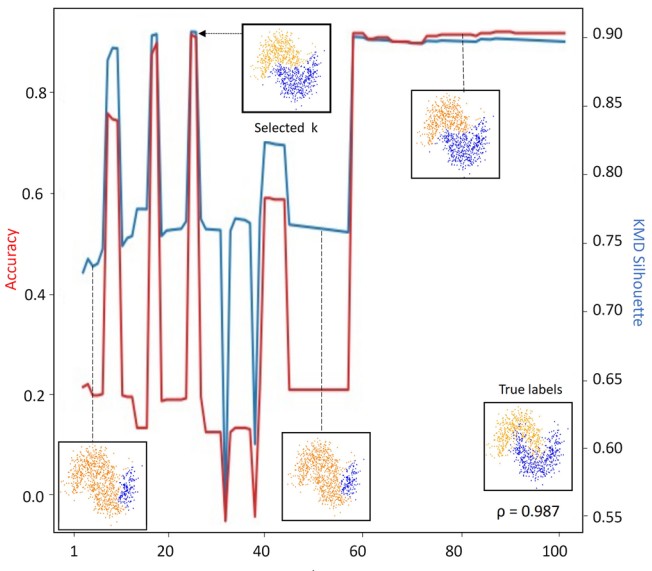

**Fig. 2 Selection of hyperparameter k.** KMD clustering accuracy (red line) and KMD silhouette score (blue line) on the high-noise half moons dataset across a range of *k* values (1–100). Insets show true labels as well as cluster assignments at *k* values of 1, 40, 50 and 80. Pearson correlation between the clustering accuracy and KMD silhouette score is shown in the bottom right corner.

confidence score such that 0.5 is the lowest confidence assignment and 1 is the highest confidence assignment (Fig. S2).

**Estimation of hyperparameter k by KMD silhouette**. Due to the drawbacks associated with cryptic hyperparameters, we asked whether it is possible to eliminate the main hyperparameter *k*, which we introduced as part of KMD linkage. We first examined the effect of the hyperparameter *k* on our clustering method. The hyperparameter *k* plays a vital role in the method, as small *k* values increase sensitivity to noise and large *k* values favor globular clusters. While testing the clustering accuracy for different datasets across different values of *k*, we found that often neither $k = 1$ (single linkage) nor $k \gg n$ (average linkage) give the best solution, and that clustering performance can vary dramatically for different values of *k* (e.g. high-noise half moons dataset, see "Evaluation on simulated datasets" for dataset details, Fig. 2). Thus, it would be difficult for a user to prespecify *k* for a new dataset.

In order to eliminate the hyperparameter *k*, we asked whether there exists some intrinsic property which may indicate the quality of a clustering solution, thus allowing hyperparameter *k* to be chosen automatically. A common method for measuring the quality of a clustering solution is the silhouette score, which is based on comparing each object's intercluster vs intracluster distances[22]. However, due to its similarity to the average linkage calculation, the silhouette score tends to favor globular clusters and thus will generally not be indicative of clustering performance when clusters are non-globular. To overcome this limitation, we propose a generalized modified form of the silhouette score which naturally matches the KMD linkage. This new generalized function, which we refer to as *KMD silhouette*, calculates the difference between an object's intercluster and intracluster distances by only using KMD linkage rather than average linkae, where *k* is selected to match the same *k* value used for clustering. In addition, we add a factor that penalizes large values of *k* (see Methods). Thus, the formula for calculating the KMD silhouette changes for different values of *k* to match the changes in the linkage function. Next, we tested whether the intrinsic KMD silhouette measure that we

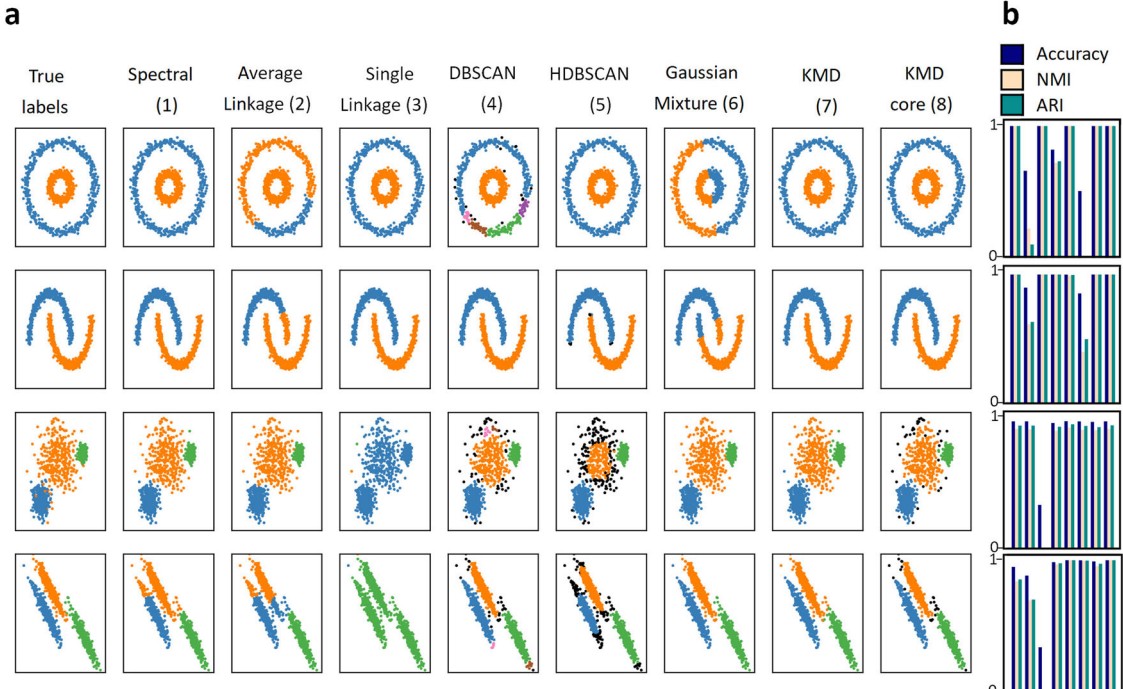

**Fig. 3 Evaluation on simulated datasets. a** Comparison of clustering algorithm performance on standard scikit-learn simulated datasets (top to bottom: nested circles, half moons, globular clusters, anisotropic clusters). Algorithms (left to right): Spectral clustering (1), hierarchical average linkage (2), single linkage (3), DBSCAN (4), HDBSCAN (5), gaussian mixture (6), KMD clustering (7) and KMD clustering core clusters (outliers shown in black) (8). Datasets (top-down): nested circles, half moons, globular clusters and anisotropic clusters. **b** Evaluation of algorithms by accuracy (blue), Normalized Mutual Information (light pink), Adjusted Rand Index (green).

proposed is predictive of the actual clustering performance when compared to ground truth, across different values of $k$ (e.g. high-noise half moons, Fig. 2). Testing this on 50 different instances of high-noise half moons, we find that KMD silhouette is predictive of all three performance metrics (median accuracy Pearson correlation = 0.89, median NMI Pearson correlation = 0.85, median ARI Pearson correlation = 0.87). Thus, we eliminate hyperparameter $k$ by running the clustering in parallel over several values of $k$ and picking the $k$ value that has the highest KMD silhouette score.

**Evaluation on simulated datasets**. We first sought to evaluate and characterize our method's performance on a standard set of simulated datasets, provided by the python package scikit-learn[23]. The scikit-learn datasets consist of five synthetic two-dimensional clustering problems (nested circles, half moons, globular clusters, and anisotropic clusters; each containing 1000 datapoints), which are generated from a mathematical function with a parameter controlling the amount of variance/noise. While these datasets are considerably different than experimental datasets, they are useful in broadly characterizing the strengths and weaknesses of standard general-purpose clustering algorithms. Importantly, in this data true cluster labels are known, allowing quantitative evaluation of clustering performance. For each clustering problem, we ran six different standard general-purpose clustering algorithms (spectral clustering, average linkage hierarchical, single linkage hierarchical, DBSCAN, HDBSCAN and gaussian mixture), as well as KMD clustering. We quantified performance using three standard performance metrics: accuracy, Normalized Mutual Information (NMI) and Adjusted Rand Index (ARI), in which the highest achievable score is 1 (see Methods for details). We did not penalize algorithms that do not classify all object, even though this likely boosts their score because the unclassified outliers are often most difficult to classify. Comparing performance across all datasets, we find that in each of the clustering problems, KMD

clustering is either highly competitive or outperforms the other approaches (Fig. 3, Table S1). KMD clustering achieves high performance scores on all datasets (nested circles: Accuracy = 1/ NMI = 1/ARI = 1, half moons: 1/1/1, globular clusters: 0.979/ 0.915/0.938, anisotropic clusters: 0.991/0.955/0.973). HDBSCAN also performs well on all datasets, but classifies 16% of the globular clusters and 11% of anisotropic clusters as outliers. The remaining clustering approaches each show a qualitative performance weakness in at least one of the datasets. We asked whether the KMD core clusters are identified with higher accuracy than the detected outliers, and indeed we observed that in all cases the performance metrics are higher when ignoring detected outliers, validating our outlier detection method, that correctly excludes irregular data objects. Taken together, we find that KMD clustering performs well on all tested clustering problems.

As one of our goals was to develop a method that performs robustly in noisy clustering problems, we next tested our algorithm on the scikit-learn datasets with a high degree of noise added, such that in some cases the clusters were even difficult to resolve visually (Fig. 4). We then reevaluated each of the clustering algorithms on these noisy datasets, and also compared to five additional clustering methods (complete linkage hierarchical, Ward linkage hierarchical, minimax linkage hierarchical, Scanpy Louvain, Scanpy Leiden) (Tables S2–S4). Remarkably, we find that KMD clustering performed well across the board (accuracy - nested circles: 0.989; half moons: 0.933; globular clusters: 0.909; anisotropic clusters: 0.992). KMD clustering was the only method that correctly captures the noisy nested circles. The second-best algorithm was gaussian mixture, which identified gaussian cluster shapes but unsurprisingly failed to capture irregular cluster shapes (accuracy - globular clusters: 0.923, anisotropic clusters: 0.996; nested circles: 0.555, half moons: 0.834). Taken together, we find that our method is the only method amongst those tested, which performed well on all four noisy clustering problems. We conclude that the KMD algorithm

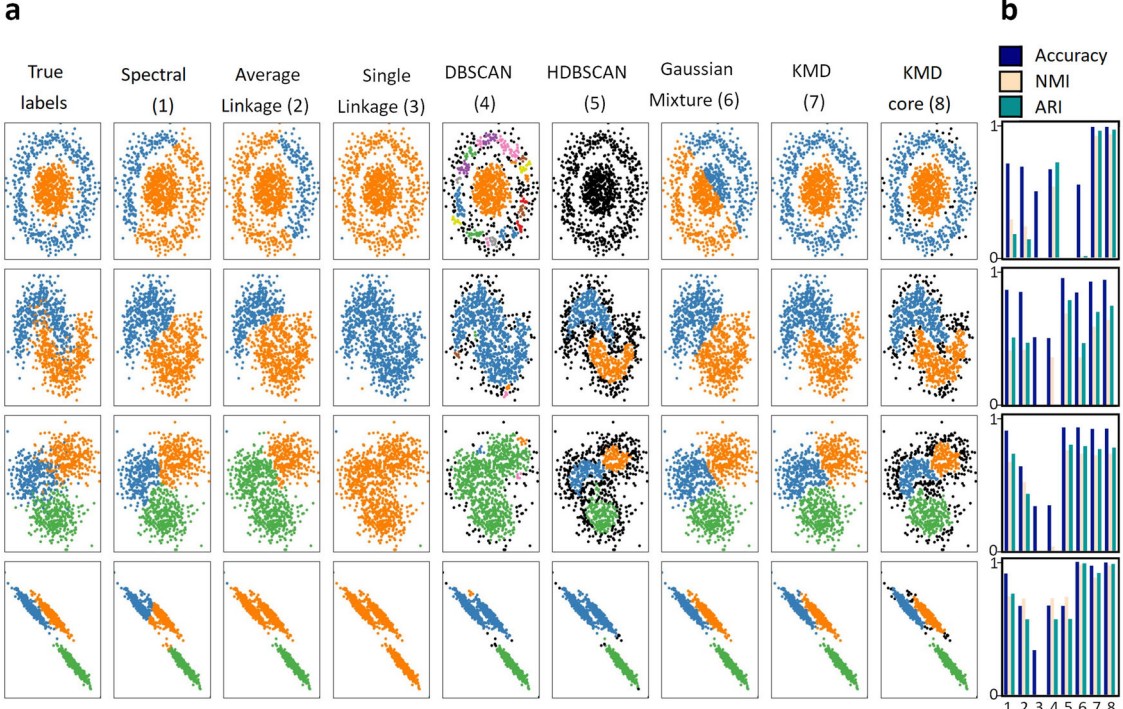

**Fig. 4 Evaluation on simulated high noise datasets. a** Comparison of clustering algorithm performance on standard scikit-learn simulated datasets with added high noise (see Methods for details). Algorithms (left to right): Spectral clustering (1), average linkage (2), single linkage (3), DBSCAN (4), HDBSCAN (5), gaussian mixture (6), KMD clustering (7) and KMD clustering core clusters (outliers shown in black) (8). Datasets (top-down): nested circles, half moons, globular clusters and anisotropic clusters. **b** Evaluation of algorithms by accuracy (blue), Normalized Mutual Information (light pink), Adjusted Rand Index (green).

consistently outperformed standard general clustering algorithms across different simulated low-dimensional datasets, including in high noise scenarios.

**Evaluation on mass cytometry data.** Next, we asked how KMD clustering will perform on complex high-dimensional biological data. To this end, we considered the problem of clustering mass cytometry data. In mass cytometry, dozens of molecular markers are measured for each cell within a sample, and we seek to identify clusters representing cell populations. To assess the performance of KMD clustering in an unbiased manner, we followed the work of Liu et al.[10], which benchmarked several state-of-the-art clustering algorithms. Liu et al.[10] use three bone marrow mass cytometry datasets for which the true labels are known from manual gating, and repeatedly ($n = 10$) sample 20,000 cells from each of these datasets (Levine15_13: 20,000 cells out of 167,044, 13 markers, 24 clusters; Levine15_32: 20,000 cells out of 265,627, 32 markers, 14 clusters; Samusik16: 20,000 cells out of 86,864, 44 markers, 24 clusters). We then compared the performance of KMD clustering to leading algorithms that were specifically designed for clustering mass cytometry data Xshift, DEPECHE, Accense, FlowSOM as well as the general clustering algorithms kmeans and PhenoGraph (Fig. 5, Table S5). In order to ensure consistency and correct usage of the algorithms, we did not rerun the algorithms but took the performance results directly from Liu et al.[10]. On the Levine15_32 dataset, we find that KMD clustering outperforms all other tested clustering methods (KMD: 0.948 accuracy, 0.944 NMI, 0.971 ARI; DEPECHE (second best): 0.892 accuracy, 0.842 NMI, 0.927 ARI). On the Levine15_13 dataset, we also find that KMD clustering performs well (ranked third) but the best is PhenoGraph (KMD: 0.812 accuracy, 0.807 NMI, 0.795 ARI; PhenoGraph: 0.918 accuracy, 0.883 NMI, 0.927 ARI). On the Samusik16 dataset, KMD clustering also performs

well (ranked second) but is slightly outperformed by PhenoGraph (KMD: 0.922 accuracy, 0.887 NMI, 0.909 ARI; PhenoGraph: 0.924 accuracy, 0.899 NMI, 0.925 ARI). Considering averaged performance across all three datasets, KMD clustering performs best in all metrics (KMD: 0.894 avg. accuracy, 0.879 avg. NMI, 0.892 avg. ARI; PhenoGraph 0.834 avg. accuracy, 0.848 avg. NMI, 0.841 avg. ARI). Overall, we find that in contrast to competing algorithms which tend to perform poorly on at least one of the datasets, KMD clustering achieves consistently high performance on each of the datasets.

**Evaluation on single cell RNAseq data.** We next asked how KMD clustering performs on extremely high-dimensional biological data, in which the number of clustered objects is much smaller than the dimensionality. Unsupervised clustering is often used on single cell RNA sequencing data in order to classify cell types based on transcriptional similarity, but these datasets can be challenging to cluster due to the sparse and noisy nature of the data[7]. To assess the performance of KMD clustering, we used three datasets containing gold standard labels determined by known markers: Lawlor17[24] (638 cells, 19927 genes, 9 clusters) containing pancreas cells, Zeisel15[25] (3005 cells, 2000 genes, 8 clusters) containing cortex and hippocampus cells and Li1[26] (561 cells, 57241 genes, 7 clusters) containing human colorectal tumor cells. We note these datasets are relatively small, and are often considered difficult to classify[27].

In clustering of scRNA-Seq data, as well as in other use cases, the number of clusters may not be known a priori, and it may be of interest to estimate this number (as done by other methods). To this end, we implemented an optimized calculation of the Density Based Clustering Validation (DBCV) measure[28], which is used to evaluate clustering solutions without assuming globular cluster shapes. We then estimated the cluster number by selecting the KMD clustering solution that maximized the DBCV (see Methods).

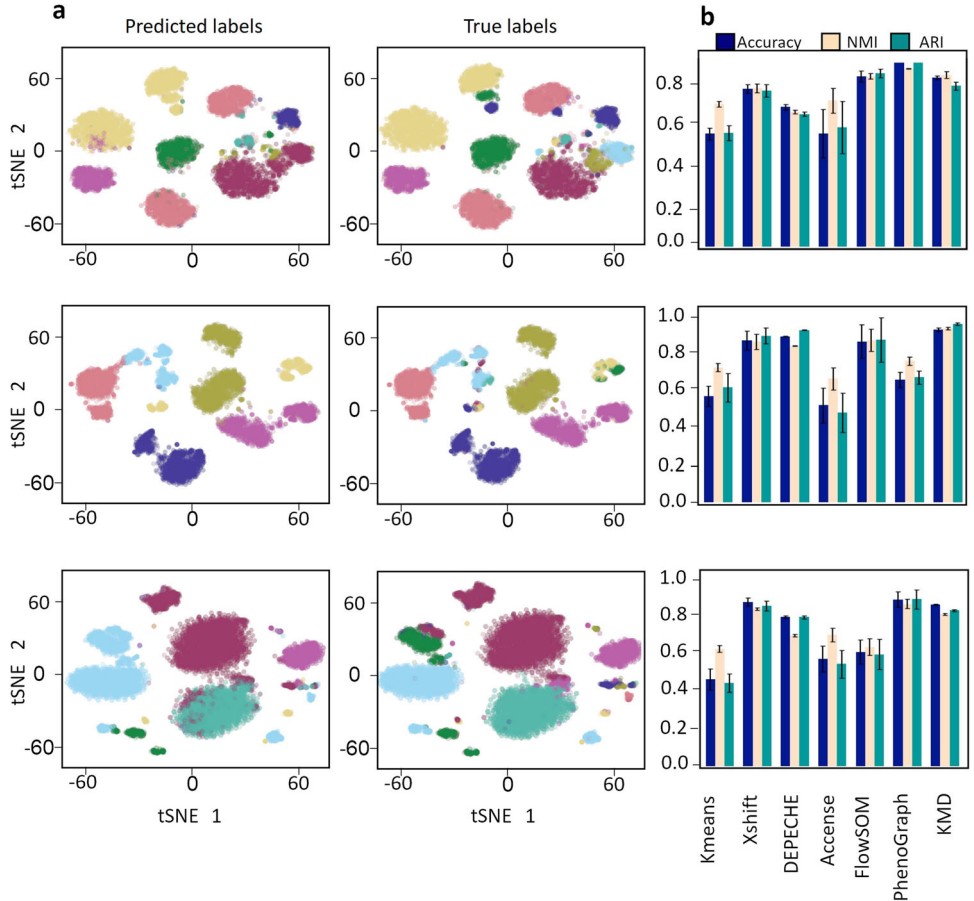

**Fig. 5 Evaluation on mass cytometry data. a** tSNE representation of mass cytometry data with clusters colored according to cell type predicted labels (left) and true labels (right) of three datasets: Levine15_13 (20,000 out of 167,044 Cells, 13 markers, 14 clusters), Levine15_32 (20,000 out of 265,627 Cells, 32 markers, 24 clusters), Samusik16 (20,000 out of 86,864 Cells, 44 markers, 24 clusters). **b** Average performance of six clustering algorithms (kmeans, Xshift, DEPECHE, Accense, FlowSOM, PhenoGraph) and KMD clustering by accuracy (blue), Normalized Mutual Information (light pink), Adjusted Rand Index (green). Error bars represent standard deviation.

We then compared the performance of KMD clustering (with cluster number estimation) to four methods that were designed for clustering scRNA-seq data: SCCAF, Scanpy Louvain, Scanpy Leiden and Seurat (Fig. 6, Table S6). In all single cell datasets, in line with current practices, we reduced dimensionality to 50 with PCA and used correlation rather than Euclidean distance as recommended in previous studies for this type of data[29]. On the Lawlor17 dataset, KMD clustering outperformed all other clustering algorithms (KMD: 0.893 accuracy, 0.790 NMI, 0.831 ARI; SCCAF (second best): 0.808 accuracy, 0.760 NMI, 0.0.769 ARI). On the Zeisel15 dataset, KMD clustering was comparable to best (KMD: 0.738 accuracy, 0.686 NMI, 0.523 ARI; SCCAF: 0.711 accuracy, 0.731 NMI, 0.586 ARI). On the Li17 dataset, KMD clustering outperformed all other clustering algorithms (KMD: 0.838 accuracy, 0.821 NMI, 0.703 ARI; Louvain (second best): 0.729 accuracy, 0.764 NMI, 0.592 ARI). KMD clustering also outperformed the competing methods on a previously suggested simulated scRNA-Seq benchmark dataset[30] (Table S7). We conclude that KMD clustering, with cluster number estimation, achieves excellent performance compared to standard scRNA-Seq clustering methods on these datasets.

**Scaling to large datasets**. While we found that KMD clustering works well on datasets containing tens of thousands of cells, we asked whether the algorithm can be scaled to much larger datasets. Indeed, recent advances in single-cell measurement

technologies have produced datasets as large as a million cells[31], posing a significant challenge for current clustering algorithms in terms of clustering performance, run time, and memory. To scale KMD clustering to large datasets, we utilized our outlier classification scheme, which classifies individual objects based on their distance to the core clusters, where distance is measured by KMD linkage with the optimal $k$ found during clustering. Given a large dataset, we randomly select a small subset of objects, run standard KMD clustering on these objects, and assign the remaining unsampled objects to clusters using the outlier classification scheme. We note that this classification scheme is very efficient in terms of computation time and memory, and thus scales well to very large datasets. We first evaluated the sampled KMD clustering approach on the full Levine15_32 mass cytometry dataset which contains 104,184 assigned cells, 32 markers and 24 clusters for which the true labels are known. To test the effect of sample size on the performance, we repeated the clustering 5 times for different sample sizes and compared the results to the true labels. Interestingly, we found that sampling just 5000 was sufficient to achieve optimal clustering performance (0.939 accuracy, 0.936 NMI, 0.966 ARI) (Figs. 7a, b, d and S3). In fact, the performance slightly decreased with sample sizes greater than 5000, perhaps suggesting that the sampling overcomes some noise effects. Next, we evaluated sampled KMD clustering on a medium-sized single-cell RNA-seq dataset (PBMC68k_Pure 62,517 cells, 17,786 genes, 6 clusters) which was previously used by Brown et al.[32] and for

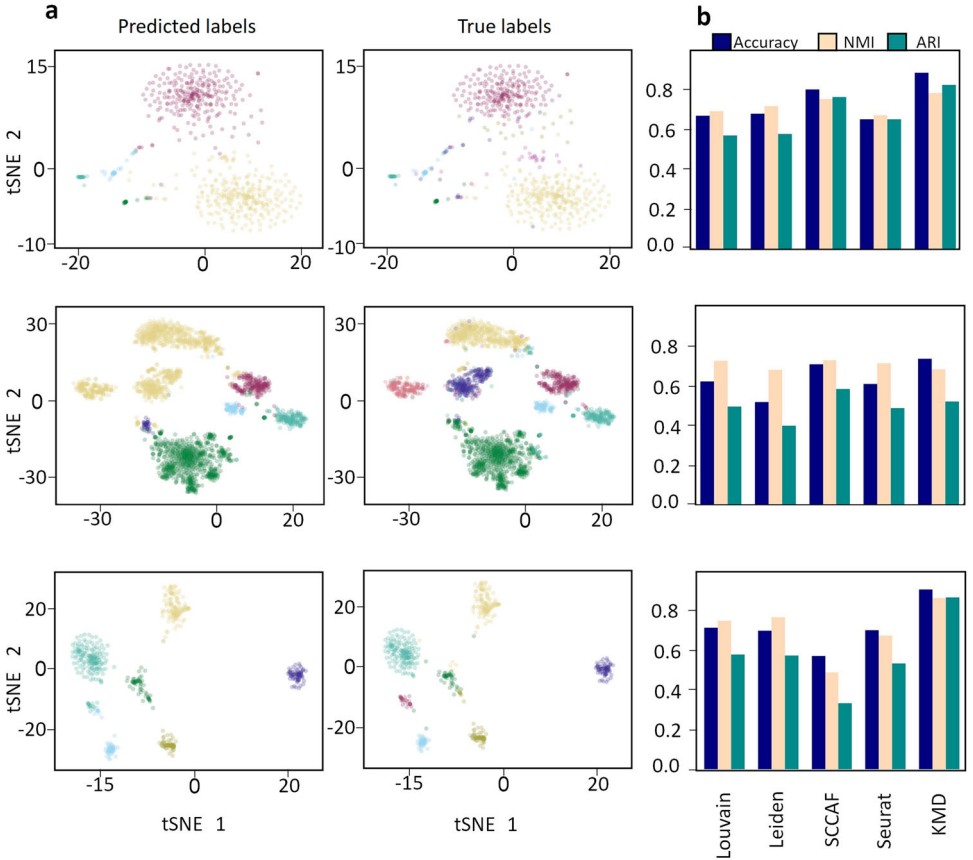

**Fig. 6 Evaluation on scRNA-seq datasets. a** tSNE representation of single cell transcriptomics data with clusters colored according to cell type predicted labels (left) and true labels (right) of three datasets: Lawlor17 (638 cells,19,927 genes,9 clusters), Zeisel15 (3005 cells, 2000 genes, 8 clusters) and Li17 (630 cells, 57,241 genes, 7 clusters). **b** Performance of four clustering algorithms (Scanpy Louvain, Scanpy Leiden, SCCAF, Seurat) and KMD clustering accuracy (blue), Normalized Mutual Information (light pink), Adjusted Rand Index (green).

which true labels are from experimental cell-type specific isolation (see Methods). Here, we sampled 5000 cells (7.3% of total). As evident from the tSNE plot, the clusters seem somewhat intermingled, and we thus expected relatively low performance from all clustering algorithms (Fig. 7b). Indeed, while the overall performance of all algorithms was low, sampled KMD clustering still performed best (KMD: 0.648 accuracy, 0.581 NMI, 0.468 ARI; Louvain: 0.577 accuracy, 0.541 NMI, 0.488 ARI) (Fig. 7e). Finally, we sought to evaluate sampled KMD clustering on a much larger scale of a million objects. While a few such datasets are available, they do not offer gold-standard cluster labels. Thus, we used Splatter[33], a tool frequently used to simulate large scRNA-seq datasets, to create a simulated dataset with 1,000,000 cells, 16,508 genes and 8 clusters. We performed sampled KMD clustering with a sample of 5000 cells (0.5% of total). We note that running KMD clustering without the sampling approach would be infeasible (see algorithm timings for full and sampled approaches in Fig. S4). We find that all algorithms performed well on the simulated dataset, with sampled KMD clustering scoring slightly higher (KMD: 0.956 accuracy, 0.876 NMI, 0.903 ARI; Louvain: 0.951 accuracy, 0.864 NMI, 0.903 ARI) (Fig. 7c, f). In conclusion, we found that a sampling-based extension of KMD clustering can be used to significantly reduce required run time and memory while maintaining high performance.

## Discussion

Average linkage and single linkage hierarchical clustering are well-established general-purpose clustering algorithms. Although they share the same framework, the difference in the definition of

linkage often leads to dramatically different results from these two algorithms[18]. Although it is well known that in machine learning there is no free lunch, we attempted to capture the best properties of both linkages by proposing a novel generalized linkage called KMD linkage. The main challenge arising from this strategy is that it introduces a cryptic hyperparameter $k$, i.e. the number of minimal intercluster distances to average when calculating pairwise cluster distance (linkage). In unsupervised learning settings, cryptic uninterpretable hyperparameters pose a major hurdle[7], because the inability to correctly estimate them a priori often leads to post hoc fitting and thus to hidden overfitting. We were thus motivated to find a way of effectively eliminating this hyperparameter by using some kind of intrinsic function which would be predictive of clustering performance without using any external labels. The silhouette score is often used to evaluate and compare clustering solutions, yet it is biased towards globular cluster shapes and is thus not adequate. We speculated that this tendency towards globular clusters may arise from the fact that in the silhouette score, the distance from an object to a cluster is calculated as the average of all pairwise distances[22], which is effectively average linkage. Thus, we considered a modified function, which we call KMD silhouette, where the distance from an object to a cluster is calculated as the $k$ minimal distances (KMD linkage), and the value of k is set to match the value of $k$ used to calculate the KMD linkage. This may seem odd, since it means that the way of scoring the clustering results is different for each value of $k$ and that two $k$ values which yield the same clustering results might be assigned different scores. We also added a term that penalizes large $k$ values, since larger $k$ values artificially inflate the score. Remarkably, we find that our

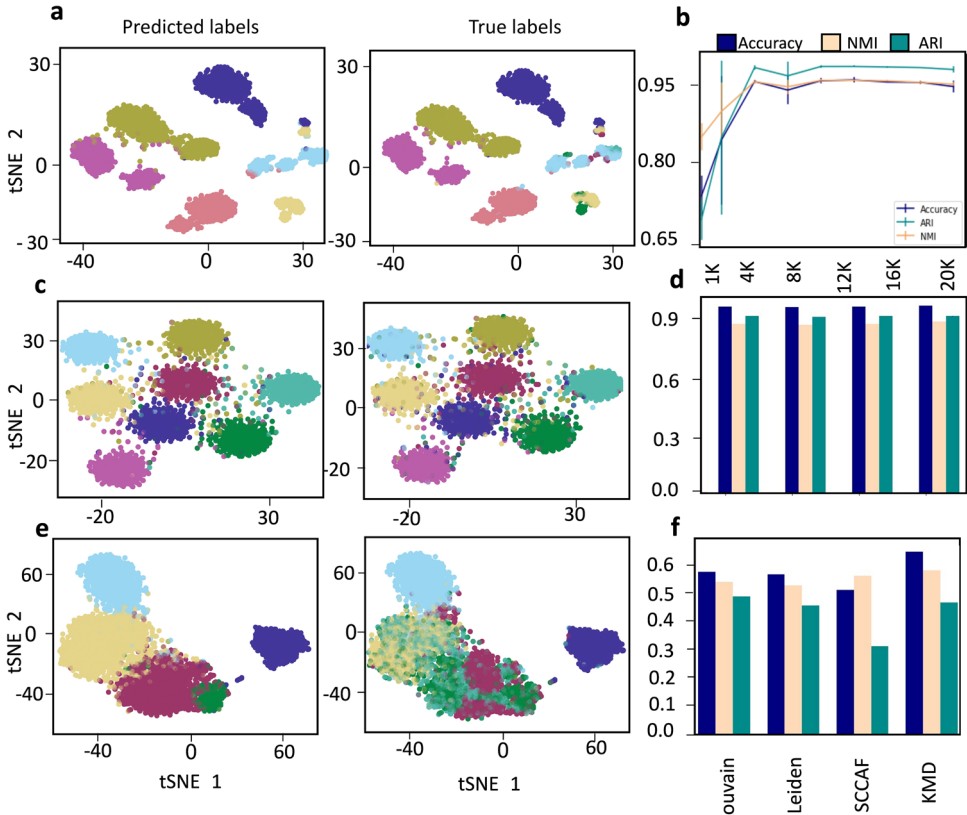

**Fig. 7 Evaluation of KMD clustering extension to large datasets. a–c** tSNE representation of scRNA-Seq data with clusters colored according to cell type predicted labels (left) and true labels (right) of three datasets: Levine15_32 (104,184 cells, 32 genes, 24 clusters), PBMC68k_Pure (62,517 cells, 32,738 genes, 6 clusters) and simZeisel15 (1,000,000 cells, 16,508 genes, 8 clusters). **d** KMD clustering ARI, NMI, accuracy score of whole dataset after cluster inferring step on the core dataset, on the Levine32dim dataset across a range of subsampled dataset size (1000–20000). **e, f** Performance of three clustering algorithms (Scanpy Louvain, Scanpy Leiden, SCCAF) and sampling-based KMD clustering accuracy (blue), Normalized Mutual Information (light pink), Adjusted Rand Index (green).

proposed KMD silhouette function predicts clustering performance across $k$ values very well. Given these results, the main hyperparameter $k$ is effectively eliminated by running the clustering across a range of $k$ values (in all cases we limited $k < 100$) and automatically selecting the solution with the highest KMD silhouette score.

The clustering performance of advanced or specialized clustering algorithms, all of which have cryptic hyperparameters (often multiple ones), can vary greatly even on data of the same type[1,7,10]. For example, results from the benchmark study by Liu et al.[10] on mass cytometry data, which we utilize in the current paper, show that the best tested algorithm in each mass cytometry dataset typically performed poorly on some other mass cytometry dataset. We speculate that the variation in the performance of other algorithms may be in part due to post hoc over-tuning of cryptic hyperparameters. In line with this, KMD clustering showed strong performance across several datasets of different types, even though it was not specifically customized for them. This may also suggest the utility of KMD clustering for new data types for which customized algorithms do not exist.

Noise and outliers in data pose a challenge for clustering algorithms, especially in biological data. While single linkage is especially prone to noise due to its calculation being based on a single distance[17], average linkage is also relatively sensitive to noise as a single incorrect merge of an outlier can have fatal consequences in later clustering stages[17]. Indeed, some clustering algorithms such as DBSCAN and HDBSCAN have been specifically designed to deal with the challenges of noisy data[21,34]. In addition to our new linkage strategy in automatic parameter selection, we use a simple scheme for dealing with outliers, based on the assumption that tiny clusters which are smaller than a size threshold can be considered to be outliers. While the outlier detection scheme is similar to that used in HDBSCAN, outlier re-assignment is based on KMD linkage matching the optimal $k$ value found during clustering. We find that KMD clustering performs extremely well with noisy data, outperforming algorithms such as DBSCAN and HDBSCAN in scenarios where noise was added to such a level that it is even difficult to discern the clusters visually in two dimensions.

While in some clustering applications the number of clusters is known in advance, in other cases the user may not have sufficient prior knowledge to estimate this number. Indeed, some clustering algorithms which we compared to do not take the number of clusters as input and instead estimate the number of clusters de novo, and this could potentially affect their prediction accuracy (although in some cases this could also increase performance metrics). While KMD clustering in general allows the user to determine the number of clusters, we also briefly explored de novo estimation of cluster numbers based on the density measure DBCV[28]. The initial results seem potentially promising based on this strategy, but further research is needed to improve cluster number estimation.

While KMD clustering performed well on the tested datasets, we note a number of computational limitations of our approach. First, we note that automatic selection of the hyperparameter $k$ requires running the clustering with a large range of $k$ values and

selecting the solution with the best KMD silhouette score. While these can be run in parallel, it does require more computational resources than a typical simple clustering approach such as standard average linkage clustering. Second, our current implementation is in python/scipy. This makes the code more accessible to users, but results in longer running times than widely used clustering algorithms that are implemented with non-interpreted languages and carefully optimized. Finally, even when disregarding language of implementation, a single run of KMD clustering may be inherently slower than some other approaches (such as single and average linkage clustering) as it requires maintaining for each pairs of clusters a list of the $k$ minimal distances between them. Due to these limitations, we proposed a sampling-based extension to our approach, which makes it easily scalable to modern datasets containing millions of objects. One limitation of this sampling approach is that it may miss very small clusters, in cases where they are not sampled well. A possible solution to this is substituting the sampling scheme. Rather than using uniform sampling, one could use a structurally-based sampling approach such as submodular optimization sampling to obtain a more representative core[35].

In conclusion, KMD clustering is a general-purpose clustering algorithm combining a novel generalized linkage function, automatic hyperparameter selection using an intrinsic function and outlier-aware partitioning. Compared to a set of tested general-purpose and specialized algorithms, KMD clustering exhibits excellent performance across a variety of datasets including extreme noise and high dimension, without dataset-specific tuning of cryptic hyperparameters.

## Methods

**KMD clustering algorithm**. The input to the KMD clustering algorithm is a $nxm$ matrix (2d ndarray) where $n$ is the number of objects and $m$ is the object dimensionality. Based on the input matrix, a $nxn$ symmetric distance matrix (2d ndarray) $D$ between all pairs of objects is computed. The default distance metric is the Euclidean norm. Next, the pairwise distances are used to populate a heap that can efficiently pop the two nearest clusters. Finally, a symmetric matrix $A$ containing the k minimal distances between every pair of clusters is initialized. $A$ is a $nxn$ matrix (2d ndarray) of pointers to lists and is initialized such that $A_{i,j}$ is a list containing the distance between objects $i$ and $j$. Output is stored in array $Z$.

Every iteration of the algorithm includes the following:

1. $x, y = $ get_minimum($heap, D$) # Return indices of two nearest clusters
2. $A_{x,*} = A_{*,x} = $ merge($A_{x,*}, A_{y,*}$) # Construct the k minimal distances of the new cluster to every other cluster by merging the respective k minimal distances of x and y to every other cluster in order. Replace cluster x with the new cluster.
3. $D_{x,*} = D_{*,x} = $ mean($A_{x,*}$) # Update distance matrix
4. $D_{y,*} = D_{*,y} = A_{y,*} = A_{*,y} = $ null # Eliminate cluster y
5. Z.append([x, y, dist(x,y)])
6. $heap$.update($A_{x,*}$) # Update the heap with the new distances

The heap implementation is a modification of the Generic_Linkage algorithm described in ref. [36] in which candidates for nearest neighbors of clusters are maintained in a priority queue to speed up the search for the two nearest clusters.

Step 2 is a critical step since in general, updating the distances of a new cluster with an arbitrary linkage can be very inefficient. However, with KMD linkage an efficient update can be performed in O($nk$). To see this, consider finding KMD($z,i$) - the KMD linkage between a new cluster $z$, which was created by merging

$x$ and $y$, and an existing cluster $i$. Since KMD($z,i$) must be a subset of KMD($x,i$) ∪ KMD($y,i$), we only need to search $2k$ distances. Furthermore, if we start with lists of length 1 and update by merge-sorting the lists and taking the minimal k elements, the update will only take O($k$) for a pair of clusters, and therefore O($nk$) for all clusters.

In addition, naively one might expect matrix $A$ to require O($n^2k$) space, since we must maintain the k minimal distances for all pairs of clusters. However, we are able to avoid this with the aforementioned implementation of $A$ as an array of list pointers. To see this, consider that initially $A$ holds only lists of length 1, and thus requires O($n^2$) space. At each iteration, rows $A_{x,*}$ and $A_{y,*}$ are merged as explained above to create a new row, and then rows $A_{x,*}$ and $A_{y,*}$ are effectively eliminated. Since the new row cannot require more memory than the union of rows $A_{x,*}$ and $A_{y,*}$, the required space cannot surpass O($n^2$), and actually decreases during the process. This type of implementation is critical, as it allows running the algorithm with large k values, which would be impractical if one was to maintain an array of size O($n^2k$).

**Outlier-aware partitioning**. Once all clusters are merged, the resulting binary tree is used to apply outlier-aware partitioning. We start with parameters $c$ (the number of clusters) and $m$ (the minimal cluster size / outlier size threshold). Starting from the root of the tree, we first find the first $c$-1 merges where both clusters have a size of at least $m$. The $c$ clusters included at these merges are defined as core clusters. Any objects which are not present in the core clusters are defined as outliers. Note that in general this partitioning is not a fixed-height tree cut as the selected merges can occur at different heights. As shown in Fig. S1, clustering results are typically robust to the choice of this parameter. In general, $m$ can be chosen to be slightly smaller than the size of the smallest expected cluster. If no a priori information is available, we suggest setting $m = $ max(2, $n/(10*c)$).

In order to assign each outlier to one of the core clusters, we calculate the KMD linkage of the outlier to each of the core clusters and assign it to the core cluster with the minimal distance. In order to provide a confidence level to this assignment, we define a confidence score that compares the KMD distance between the outlier $v$ and its two nearest core clusters $C_1$ (nearest cluster) and $C_2$ (second nearest cluster):

$$confidence(v) = 1 - \frac{dist(v, C_1)}{dist(v, C_1) + dist(v, C_2)} \qquad (1)$$

The confidence score ranges between 0.5 (lowest confidence) to 1 (highest confidence). all comparisons of clustering performance in the paper were done using these outlier assignments, such that no objects were left out unless stated otherwise.

**KMD silhouette**. The KMD silhouette is an intrinsic function, inspired by the silhouette function, that is used to assess the quality of a clustering solution. The core component of the silhouette is the difference between $a_i$, the average distance of object $i$ to the other objects of the assigned (nearest) cluster, and $b_i$, the average distance of object $i$ to the objects of the second nearest cluster. Let us denote this difference $d_i = b_i - a_i$. In KMD silhouette we use the same quantity $d_i$, except that we define $a_i$ and $b_i$ slightly differently by using the average of the $k$ minimal distances rather than all distances. For every run $t$ of the clustering algorithm with a $k$ value of $k_t$, we calculate the average of these differences and denote this as $s_t$, i.e. $s_t = \frac{1}{n}\sum_i d_i^t$. Finally, the KMD

silhouette of run $t$ is defined as:

$$KMDsilhouette(t) = \sqrt{\frac{s_t - \min_i(s_i)}{\max_i(s_i) - \min_i(s_i)}} - \frac{k_t}{n} \qquad (2)$$

**Clustering evaluation.** In order to perform clustering evaluation in an unbiased manner, we chose to work only with datasets where true labels are known. Following Liu et al., we used three different performance metrics: Accuracy, Normalized Mutual Information (NMI) and Adjusted Rand Index (ARI). We denote the true and predicted labels as vectors of integers $t$ and $p$ in which the $i$-th element represents the true and predicted cluster, respectively.

*Accuracy.* Given a one-to-one matching between predicted and assigned clusters, the accuracy is defined as

$$Accuracy(p, t) = \frac{1}{n} \sum \mathbf{1}_{p_i = t_i} \qquad (3)$$

where $1_{p_i = t_i}$ is an indicator function that counts when $p_i = t_i$. The optimal one-to-one matching that maximizes accuracy was found by using the Hungarian algorithm[37].

*Normalized Mutual Information (NMI).* We defined mutual information of t and p as:

$$I(t, p) = \sum_{p,t} P_{tp}(t, p) \log(P_{tp}(t, p) / p_p(p) p_t(t)) \qquad (4)$$

Where $p_t(t)$ and $p_p(p)$ are the probability distributions and $P_{tp}(t, p)$ is the joint distribution

NMI is the more commonly used normalized form:

$$NMI = \left( \frac{2I(p, t)}{H(t) + H(p)} \right) \qquad (5)$$

Where $H(t)$, $H(p)$ are the information entropies. NMI is large if $p$ is an optimal clustering result. $t = p$ corresponds to NMI = 1.

**Adjusted Rand Index (ARI).** Rand index (RI) can be computed using the following formula:

$$RI = \frac{TP + TN}{TP + FP + FN + TN} \qquad (6)$$

*TP* - objects in a pair are placed in the same group in $t$ and in the same group in $p$. *FP* - objects in a pair are placed in the same group in $t$ and in different groups in $p$. *FN* - objects in a pair are placed in the same group in $p$ and in different groups in $t$. *TN* - objects in a pair are placed in different groups in $t$ and in different groups in $p$.

ARI is calculated by adjusting RI using the following scheme:

$$ARI = \frac{RI - Expected\_RI}{1 - Expected\_RI} \qquad (7)$$

Where the *Expected_RI* is defined as:

$$Expected\_RI = \frac{(TP + FP)(TP + FN) + (TN + FP)(TN + FN)}{(TP + FP + FN + TN)^2} \qquad (8)$$

**Evaluation on simulated datasets.** All simulated datasets were generated using scikit-learn random sample generators with 1000 objects and seed = 1. The nested circles dataset was generated using the make_circles generator with the parameters factor = 0.3 and noise = 0.05 (noise = 0.14 for high noise version). The half moons dataset was generated using the make_moons generator with noise = 0.05 (noise = 0.24 for high noise version). The anisotropic clusters dataset was generated using a transformed

dataset generated by the make_blobs generator. The transformation was the dot product of the dataset with the array [[0.6, −0.6], [−0.4, 0.8]]. The random state was set to 170 (185 for high noise version). The globular clusters dataset was generated using the make_blobs generator. The random state was set to 170 (185 for high noise version) and the standard deviation was set to [1.0,2.5,0.5] ([2.0,2.0,2.0] for high noise version). All datasets objects were standardized.

The noisy half moons dataset used in the analyses shown in Figs. 1, 2 and Supplementary Fig. 2 was generated with parameters noise = 0.24 and seed = 5 (Fig. 1) or seed = 3 (Fig. 2 and Supplementary Fig. 2).

In all cases where we did not use default running parameters for competing algorithms, we selected new parameter settings that improved the performance of those algorithms relative to the performance under default settings.

Single linkage clustering was run with the AgglomerativeClustering scikit-learn algorithm, using the non-default parameter connectivity to specify the number of neighbors to use for each dataset: 2 neighbors for nested circles and half moons, 10 neighbors for anisotropic clusters and globular clusters.

Average linkage was run with the AgglomerativeClustering scikit-learn algorithm, using default parameters.

Complete and Ward linkages were run using Scipy's cluster.hierarchy.linkage method using default parameters followed by cluster.hierarchy.fcluster with criterion of maxclust to convert the linkage matrix to cluster predictions.

Minimax linkage[38] was run using pyminimax (https://pypi.org/project/pyminimax/) with default parameters, followed by cluster.hierarchy.fcluster with criterion of maxclust to convert the linkage matrix to cluster predictions.

Spectral clustering was run with the SpectralClustering scikit-learn algorithm, using non-default parameters eigen_solver =' arpack' and affinity = 'nearest_neighbors'.

DBSCAN was run with the DBSCAN scikit learn algorithm, using the non-default parameter setting: eps = 0.15 for nested circles and half moons, eps = 0.18 for globular clusters, eps = 0.15 for anisotropic clusters.

HDBSCAN was run with the hdbscan python package[39], using the non-default parameter setting: minimal cluster size = max(2, number of objects/(10*number of clusters)).

The Scanpy Louvain and Leiden algorithms were executed with randomization seed of 0 and default parameters as follows: First, a neighborhood graph was calculated with number of neighbors = 15, then the Louvain and Leiden clusterings were computed with resolution = 1.

SCCAF was executed using the SCCAF_optimize_all function with the following parameters: The start optimization point was precomputed Louvain annotation; optimization name (prefix) was set to 'L1'; basis was set to 'tsne'. The analysis was run 10 times and scores were averaged in order to eliminate the heterogeneity of the dataset random split to train/test datasets.

Gaussian mixture was run with GaussianMixture scikit learn algorithm, using default parameters.

KMD clustering was performed as described minimal cluster size = max(2, number of objects/(10*number of clusters)).

**Evaluation on mass cytometry datasets.** The datasets Levine15_13 (167,044 Cells), Levine15_23 (265,627 cells) and Samusik16 (86,864 Cells) are known benchmarking mass cytometry datasets and have been used in previous comparisons of algorithms[8,10]. The true label annotations are known cell types that were manually gated. The transformed and filtered datasets were downloaded from the "flowrepository" repository (http://flowrepository.org/id/FR-FCM-ZZPH)[8]. Cells that were unassigned or had ambiguous

annotations were discarded. In the interest of consistency and fairness, clustering results for kmeans, Xshift, DEPECHE, Accense, FlowSOM and Phenograph were taken from ref. [8]. Following Liu et al., we randomly sampled 20,000 cells ten times from each dataset, ran KMD clustering (minimal cluster size = 50, correlation distance) and calculated the mean and standard deviation for each performance metric. K value for KMD was found by scanning in parallel from 1 to 100 with a step size of 5 and a randomization seed of 1.

**Evaluation on single cell RNA-seq datasets.** The Townes et al.[30] simulated dataset was created according to the data simulation procedure in https://github.com/willtownes/glmpca/blob/master/vignettes/glmpca.Rmd. The simulated profile is of 3 clusters, each with 50 cells and 5000 genes, 500 of which are differentially expressed between the clusters. In addition, cells are randomly marked as having a high total count or a low total count, a property that should be ignored by the clustering algorithm.

The Li17 dataset consists of 630 single cells sampled from 7 cell lines, with the expression profiling differing between patients due to intratumoral heterogeneity. The dataset was preprocessed by filtering out the 5% of highest and lowest expressed genes.

The Lawlor17 dataset consists of single cell transcriptomes of 638 human islet cells obtained from five non-diabetic and three type 2 diabetic cadaveric organ donors. The dataset was preprocessed as suggested in ref. [11]: Normalization by the natural logarithm of one plus the input array; cells with under 200 expressed genes were filtered out; genes that were expressed in less than 3 cells were filtered out; cells with ambiguous annotations were filtered; each cell was normalized by total counts by all genes.

The Zeisel15 dataset consists single cell transcriptome of 3005 mice cerebral cortex cells. The dataset was preprocessed as suggested in ref. [11]: Normalization by the natural logarithm of one plus the input array; genes that were expressed in less than 3 cells were filtered out; the 2000 genes with the highest variance were selected for the analysis.

The Louvain and Leiden algorithms were executed with default parameters as follows: First, PCA components were calculated with number of components = 50; next, neighborhood graph was calculated with number of neighbors = 15; finally, Louvain and Leiden clusterings were computed with resolution = 1.

SCCAF was executed using the SCCAF_optimize_all function with the following parameters: The start optimization point was precomputed Louvain annotation; optimization name (prefix) was set to 'L1' for Lowler17 and Li17, and 'L2' for Zeisel15; the algorithm ran on PCA data calculated as explained above, and the minimum self-projection accuracy was 0.93. The analysis was run 10 times and scores were averaged in order to eliminate the heterogeneity of the dataset random split to train/test datasets.

Seurat was run by the following procedure (all with default parameters): Preprocessed data was first scaled with ScaleData, variable genes were found with FindVariableFeatures, dimensions were reduced with RunPCA and then clustered with FindNeighbors and FindClusters.

KMD clustering was run with correlation distance as recommended in ref. [29] and with minimal cluster size of 10 due to the small sizes of the clusters in these datasets. To estimate the number of clusters, we first optimized the Python implementation of DBCV from https://github.com/christopherjenness/DBCV over 1000 fold. Our implementation is available at https://github.com/hkariti/DBCV. Next, KMD silhouette was used to find the best $k$ value for every possible number of clusters (KMD clustering identifies the possible range to be 2–14 clusters for these datasets). Finally, for each number of clusters we calculate

the DBCV (after removing outliers) of the solution with the $k$ value selected by the KMD silhouette, and choose the number of clusters maximizes the DBCV score.

**Evaluation on large datasets.** Levine_32dim was preprocessed as detailed in the mass cytometry section.

The PBMC68K_Pure dataset was taken from Brown et al.[32] and preprocessed according to their instruction as follows. Cells were first filtered according to the main_process_pure_pbmc.R script, followed by selection of 6 out of 11 populations for further analysis. The resulting data matrix was then CPT-normalized and log-transformed, the top 1000 variable genes were filtered and scaled and finally dimensionality was reduced to 20 using PCA.

The simulated dataset was created using the python implementation of Splatter[33]. Splatter estimates general parameters of a given dataset and then generates new data using those parameters. The following parameters were estimated by Splatter from the Zeisel15 dataset: seed = 1, ncells = 1000000, ngenes = 6508, ngroups = 14, libloc = 7.64, libscale = 0.78, mean_rate = 7.68, mean_shape = 0.34, expoutprob = 0.00286, expoutloc = 6.15, expoutscale = 0.49, diffexpprob = 0.025, diffexpdownprob = 0, diffexploc = 1, diffexpscale = 1, bcv_dispersion = 0.448, bcv_dof = 22.087, ndoublets = 0, nproggenes = 400, progdownprob = 0, progdeloc = 1, progdescale = 1, progcellfrac = 0.35, proggroups = [1–4], minprogusage = 0.1, maxprogusage = 0.7. After simulation, data was preprocessed using Scanpy: genes with less than 1 count were filtered out, total count of cells was normalized with sc.pp.normalize_per_cell(key_n_counts = 'n_counts_all') and the top 1000 dispersed genes were chosen. Cell counts were then normalized again, scaled and reduced to 50 dimensions using PCA.

Louvain and Leiden were run from Scanpy using neighbors = 15 and resolution = 1 (default). SCCAF was run 10 times with default basis and initialized on Louvain results. KMD clustering was run with minimal cluster size of 50, k range of 1 to 99, internal subsampling of 5000 and random state of 1.

**Reporting summary.** Further information on research design is available in the Nature Portfolio Reporting Summary linked to this article.

## Code availability
KMD clustering is implemented in python using the numpy/scipy and scikit-learn libraries and is available including at https://doi.org/10.5281/zenodo.8344742. The repository includes python code and scripts and Jupyter notebooks for reproducing shown results.

## Data availability
Generated datasets and outputs of described pipelines is available at https://doi.org/10.5281/zenodo.8345095.

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

## Acknowledgements

We thank Nir Ailon, Shai Shen Orr, Amit Zeisel and members of the Kaplan Lab for helpful discussions and comments. We thank David Cohen for server administration and maintenance. This research was funded by Israel Science Foundation Individual Research Grant 1479/18 and the Azrieli Faculty Fellows program.

## Author contributions

A.Z. and N.K. conceived the project. A.Z., H.K. and N.K. were involved in all other aspects of research.

## Competing interests

The authors declare no competing interests.
