## [Peer Review File · Communications Biology]

REVIEWERS' COMMENTS:

Reviewer #1 (Remarks to the Author):

The authors have extensively revised their manuscript and comprehensively addressed the previous set of review comments.

In particular, one of the main comments previously was that the wording in the previous version of the manuscript made excessive claims regarding the general applicability of the method (KMD clustering) to real biological datasets, compared to the evidence provided in the evaluations. This has now been comprehensively addressed with both extensions to the methodology and additional evaluations and datasets used.

The method (KMD clustering) is well-described (i.e. "k-minimal-distances linkage" clustering, which combines aspects of both single-linkage and average-linkage hierarchical clustering) and intuitive (i.e. "more robust to noise than single linkage, yet less biased towards globular clusters than average linkage" as described by the authors), and is a welcome addition to the literature on clustering algorithms. The approach to computationally select the hyperparameter k and the definition of the "KMD silhouette" measure are also useful and clearly described.

The new extension of the methodology in the updated manuscript to enable its application to large datasets with up to a million cells using a sampling-based approach is especially useful, and is also clearly explained. Runtime / scalability evaluations have now also been provided in the supplementary figures.

The additional extension regarding the "density based clustering validation (DBCv) measure" to computationally select the number of clusters is also a promising and useful addition.